# Quantifying Urban Activities Using Nodal Seismometers in a Heterogeneous Urban Space

**DOI:** 10.3390/s23031322

**Published:** 2023-01-24

**Authors:** Yunyue Elita Li, Enhedelihai Alex Nilot, Yumin Zhao, Gang Fang

**Affiliations:** 1Sustainability Geophysics Project, Department of Earth, Atmospheric, and Planetary Sciences, Purdue University, West Lafayette, IN 47907, USA; 2Sustainability Geophysics Project, Department of Civil and Environmental Engineering, National University of Singapore, Singapore 119077, Singapore

**Keywords:** urban seismology, mobility detection, COVID-19 mitigation, compliance and monitoring

## Abstract

Earth’s surface is constantly vibrating due to natural processes inside and human activities on the surface of the Earth. These vibrations form the ambient seismic fields that are measured by sensitive seismometers. Compared with natural processes, anthropogenic vibrations dominate the seismic measurements at higher frequency bands, demonstrate clear temporal and cyclic variability, and are more heterogeneous in space. Consequently, urban ambient seismic fields are a rich information source for human activity monitoring. Improving from the conventional energy-based seismic spectral analysis, we utilize advanced signal processing techniques to extract the occurrence of specific urban activities, including motor vehicle counts and runner activities, from the high-frequency ambient seismic noise. We compare the seismic energy in different frequency bands with the extracted activity intensity at different locations within a one-kilometer radius and highlight the high-resolution information in the seismic data. Our results demonstrate the intense heterogeneity in a highly developed urban space. Different sectors of urban society serve different functions and respond differently when urban life is severely disturbed by the impact of the COVID-19 pandemic in 2020. The anonymity of seismic data enabled an unprecedented spatial and temporal resolution, which potentially could be utilized by government regulators and policymakers for dynamic monitoring and urban management.

## 1. Introduction

The Earth’s surface is constantly vibrating. Although people may not notice these minute vibrations, seismometers are sensitive to measure them with very high accuracy. Conventionally, seismometers are deployed to understand natural activities, such as earthquakes, volcano eruptions, and ocean waves (e.g., [1,2]), because with large magnitude events, they become natural hazards that may cause hundreds of millions of dollars in damage and thousands of casualties (e.g., [3]). As most of these natural activities happen at low-frequency bands, conventional seismometers are designed to be sensitive to and record vibrations at frequencies below 25 Hz. Moreover, most seismometers are deployed at remote locations to avoid overwhelming human-induced vibrations around 10 Hz. Nonetheless, as human activities expand their footprints, human activities, such as nuclear explosions, fracking, and construction [4,5,6], are also clearly measured and analyzed from seismic data. Data calculated from these seismometers also demonstrate that the global seismic vibrations between 4 and 14 Hz were reduced significantly due to the lockdown measures [7,8] to curb the spread of COVID-19 in April 2020.

Recently, nodal high-frequency seismometers (up to 2000 Hz) significantly improve the availability and flexibility of seismic measurements. The cost of individual sensors and the challenges associated with their deployment are greatly reduced. When installed in an urban environment, seismometers measure signals that are dominated by the road traffic [9,10,11], and the ambient noise level correlates well with urban mobility [12,13]. However, large urban systems are extremely heterogeneous in their functions and interactions with society. Ambient seismic energies at different frequencies [14] provide an aggregated measurement of urban activities within a few kilometers range from the nodal seismometer at the intermediate frequency range (4 to 14 Hz, for example). As the frequency bands move to higher frequencies, the range of sensitivity decreases, providing more spatial resolution to urban activities. Moreover, high-frequency seismic data recorded by a single nodal seismometer can be distilled for statistics of specific yet everyday human activities, such as aircraft [15], road motor traffic [16], and urban runners [17]. These human-induced noise recordings at different locations provide a unique window to assess the dynamic activities in different urban sectors.

Since the initial outbreak of the COVID-19 pandemic in December 2019 [18], human life has been profoundly altered. Although many countries have recovered their economic activities almost fully, the adverse effects of the pandemic are still felt across the globe in many regions. At the onset of the pandemic, countries around the world adopted lockdown measures of various degrees of stringency to reduce mobility, inter-personal contact, and, hence, the reproduction rate of the virus [19]. Megacities, such as the city-state of Singapore, are home to millions of residents whose health and livelihood are most severely impacted by the pandemic [20,21,22]. Normal urban activities have clear, regular, and cyclic temporal patterns and highly heterogeneous spatial patterns based on the urban functions of each locality [23]. Therefore, a real-time understanding of how different urban sectors undergo the impact of the COVID-19 pandemic is vital for the smooth operation of a large urban system, for dynamic pandemic management, and for long-term policy making [24]. Current monitoring solutions are often based on surveillance cameras and cell phone signals, which trigger privacy concerns and resistance in adaptation [25,26]. When Google and Apple publish their mobility monitoring data with users’ consent, personal usages in cities are anonymized, aggregated, and eventually presented statistically for locations of the same function [27,28]. Consequently, the inherent heterogeneities of urban activities are not sufficiently preserved and a detailed analysis of the pandemic’s impact on different sectors of urban society is still missing from the literature.

In this study, we compare ambient seismic noise recorded across an array of nodal seismometers at different frequency bands to demonstrate the effects of seismic aggregation and resolution. From the high frequency (40 Hz and above) data, we quantify the intensity of human activities at each location and highlight the urban heterogeneity and variability even if urban blocks are placed within close proximity. The sensor deployment has minimal intrusion on the environment, and the signal analysis is entirely anonymous. Hence, we achieve unprecedented spatial and temporal resolution without incurring privacy concerns.

## 2. Materials and Methods

Like most megacities, Singapore went into a COVID-19 lockdown, called a Circuit Breaker (CB) lockdown, on 8 April 2020. With the epidemiology observation, the government implemented a two-stage opening policy on 1 June and 19 June as Phase 1 and Phase 2 openings, respectively. Target reductions of urban mobility at 70%, 50%, and 40% during these three time periods (CB, Phase 1 and 2 openings), respectively, were designed to reduce interpersonal contact. On the eve of the lockdown, we deployed a nodal seismometer array at four different locations within a two-kilometer radius of the National University of Singapore (NUS) (Figure 1a). These are Smart Solo IGU-16 3C nodal seismometers, each with three-component velocity phones, the data logger, the GPS, the battery, and the storage disk all in one compact unit (Figure 1b). The NUS hilltop site is on top of Kent Ridge, surrounded by tropical jungles with only a single-lane local road access to a limited number of laboratories. The NUS exit site is located by a four-lane road, directly receiving traffic coming in and out of the campus. The West Coast Highway is a six-lane highway, directly supporting the traffic load coming out of the Pasir Panjang Terminal and connecting with the rest of the city. The West Coast Park site is at the intersection between the Pasir Panjang Terminal and the West Coast Park, receiving the truck traffic from the terminal port and the pedestrians inside the park. At the first three sites, 3-component nodal seismometers are deployed within 1 m from the road pavement (For one example, see Figure 1c). At the last site, the seismometer is deployed on a large lawn, 7 m from the pavement. All four locations are surrounded by local roads, highways, and expressways. At each site, the N-component is aligned with the traffic direction along the road; hence, the E-component is perpendicular to the road.

Figure 2 compares the cumulative power spectral density (PSD) of the vertical component seismic recording at each site between 1 Hz and 30 Hz. The West Coast Highway site (Figure 2c) records the strongest vibration energy within this frequency band, an order of magnitude higher than the NUS exit site (Figure 2a), and the West Coast Park site (Figure 2d). The NUS hilltop site (Figure 2b) records the weakest vibration energy, nearly four orders of magnitudes weaker than the highway site. Such difference in energy levels at different sites agrees well with the local traffic condition. Nonetheless, all four sites show a remarkably similar day-night variational pattern, with energy picking up rapidly around 5 am (local time) and dying off gradually from 8 pm to midnight. There is also an overall agreeing trend of increasing energy from CB to Phase 1 and Phase 2 opening. Such similarities suggest that recordings of seismometers at the four sites reflect an aggregation of urban activities because seismic waves emitted from the distributed locations of these activities could propagate up to a few kilometers at this frequency band. Therefore, ambient seismic recording at a single seismometer between 1 Hz and 30 Hz can be used as a proxy of the average urban activity, as used by [7] in their global observations. However, extracting additional site-specific information from data within this frequency band is challenging.

In contrast, Figure 3 compares the cumulative PSD at each site between 30 Hz and 150 Hz. At this high-frequency band, ambient seismic energy is one to two orders of magnitudes weaker than at the low frequencies (Figure 2) due to the increasing attenuation of the higher-frequency seismic waves. Therefore, energy recorded within this high-frequency band is more likely emitted within a few meters of the sensor, providing more site-specific information. Figure 3 shows significantly different daily and long-time patterns at four sites. The NUS exit site (Figure 3a) is the only site maintaining the same day–night variational pattern as its low-frequency counterpart. Activity around this site picks up at 7 am and dies off at 11 pm. At the NUS hilltop site (Figure 3b), the high-frequency energy focuses on constant 8 am events throughout the monitoring period. In addition, a dozen sporadic occurrences (shown in bright yellow strips) correspond to the tropical thunderstorm events during the five-month monitoring time. To support these observations, rainfall data at the Kent Ridge weather station (denoted by the orange star in Figure 1) are plotted on top of Figure 3b. It is obvious that these large energy events occur on days with intensive tropical downpours (defined by the highest 60 min rainfall greater than the mean value of 6.3 mm). Compared to daily meteorological data, seismic data resolve the thunderstorm events with hourly resolution.

At the West Coast Highway site (Figure 3c), the high-frequency energy is almost constant from April to August 2020, except for 11–19 April and 5–19 August, where the energy is drastically lower. Knowing the site condition of a major coastal highway, it is unlikely that this decrease in energy comes from a reduced traffic volume during the corresponding period. Instead, this energy reduction is likely due to a temporary shutdown of the construction activity for an apartment complex at this site because of the COVID-19 outbreak among the construction workers. Lastly, we observe an increase in high-frequency energy at the West Coast Park site, adjacent to the Pasir Panjang Terminal Port, at least two weeks before the scheduled Phase 1 opening. This suggests an increase in port activity during the lockdown period. Although the high-frequency ambient seismic energy provides a qualitative yet straightforward measure of the highly localized urban activity, the type of the urban activity is still obscure from the energy measure itself. For example, it is essential to distinguish between the energy from the road traffic and that from the construction site, as their changes may suggest different variations in urban functions.

In this study, we quantify specific activity information from the vibration measurements of the seismometers at the high-frequency band (above 30 Hz). We unwrap the entangled information in the time domain records in its two-dimensional spectrogram representations. As each passing motor vehicle directly impacts the ground near the seismometer, its seismic signature has a short duration (about two seconds) and a wide frequency band that stands out from the background noise level [16]. Therefore, we use Hanning windows of 0.5 s with 12.5% overlap to compute the PSD of the continuous record. We then sum the energy between 30 Hz and 150 Hz and identify each energy peak as an individual motor vehicle. The peaks are defined by two parameters: the relative height of the peak (prominence) and the time interval between the two nearest peaks (isolation), which are tuned according to site conditions. There are four lanes on the roads at the NUS exit and the West Coast Park sites, but six lanes on the West Coast Highway. Therefore, the prominence and isolation parameters are tuned differently (Table 1). As an example, we illustrate the process of identifying three motor vehicles in Figure 4.

Runner identification is a more challenging task as the vibration generated by a runner is significantly smaller than that of a motor vehicle and is often buried in the ambient background noise. In addition, the frequent tropical thunderstorms in Singapore also generate strong vibration signals that could be misidentified as runners. Therefore, a more delicate signal processing workflow is applied for runner identification [17]. Here, we take advantage of the periodic feature of human footsteps because the cadence (number of steps taken per minute) for a person tends to be steady for a given activity. A runner’s cadence (150–200 steps/min) is higher than a walker’s (100–120 steps/min), for example. To extract the cadence feature from the seismic data, the horizontal component of the vibration signal is further filtered between 40 Hz and 100 Hz. The spectrogram is then calculated using a shorter Hanning window of 0.064 s with 12.5% overlap. The averaged PSD is then separated into 8-second segments and inspected in the frequency domain. Compared to random noise and motor vehicle signals, the runner’s signal has three harmonics. The primary frequency occurs in the range of 2–4 Hz corresponding to the cadence of the runner. The second and third harmonics occur at about 2 and 3 times the primary frequency, respectively, due to the discrete nature of the footsteps. Compared to thunderstorms, the runner signal has a smaller amplitude yet a larger contrast between the peak amplitude and the median amplitude within a broader frequency band.

A runner is identified on each 8-second recording, and its corresponding averaged PSD if all following conditions are satisfied:Amplitude of the filtered data in each window should be smaller than a pre-defined threshold.The spectrum of the averaged PSD should contain primary and second harmonics. The primary should occur in the frequency range of 2–4 Hz. The second harmonic should occur at around 2 times the primary frequency.Amplitude at the primary frequency of the averaged PSD should be higher than a pre-defined threshold.

Figure 5 illustrates the process of footstep identification. More details of the footstep identification algorithm can be found in [17].

We compare our results with the COVID-19 Community Mobility Reports provided by Google (reference needed). For different countries and selected regions within some countries, Google Mobility Reports provide movement changes according to different categories of places, including groups, such as parks, workplaces, and residential areas. They do not provide absolute visitor numbers, but only the percentage change in mobility with respect to baseline data that represent a normal situation before COVID-19 lockdown measures. To respect and preserve people’s privacy, user data are aggregated and anonymized to move personally identifiable information. Small data samples that prevent confidential and anonymous estimates will result in data gaps. Google Mobility Reports are provided with one-day time intervals and only one data point is provided for each category in Singapore. For more details of the Google COVID-19 Community Mobility Reports, we refer the readers to their website (accessed on 17 January 2023) https://www.google.com/covid19/mobility/.

We also compare our results with the meteorological data, particularly the precipitation data (Data downloaded from Meteorological Service Singapore (accessed on 17 January 2023) http://www.weather.gov.sg/climate-historical-daily/). The rainfall data are selected at the Kent Ridge weather station, located close to our NUS hilltop site. From the rainfall data, days with intensive tropical downpours (highest 60 min rainfall greater than the mean value 6.3 mm) are plotted in orange (Figure 3b). We highlight these days with strong precipitation events and compare them with our data.

## 3. Results

Figure 6 shows the motor vehicle count extracted from high-frequency seismic data at four sites, and the daily and hourly statistics of the motor vehicles are shown in Figure 7 for the NUS exit and the Pasir Panjang Terminal sites. Compared to the average ambient seismic energy at low frequencies (Figure 2) and high frequencies (Figure 3), the motor vehicle count at different sites shows evident localized mobility variations without interferences of other ambient noise sources.

The traffic pattern on the NUS campus (Figure 6a) represents the working routine of ordinary urban residents that starts around 8 am and finishes around 7 pm. During the lockdown period, no rush hours are observed due to the restricted access to non-essential personnel. Instead, the traffic volume is almost constant during the day hours (8:00–19:00). Yet, a strong weekday–weekend contrast in motor vehicle count is still visible. The trend of daily traffic count agreed very well with Google Mobility Report for workplaces (Figure 7a). The traffic volume exiting the NUS campus is the lowest (below 10 per hour) every day between 1:00 and 5:00. The mobility trend conforms well with the government restrictions, showing an immediate decrease after CB, a step increase after Phase 1 opening, and a more gradual increase after Phase 2 opening. In addition, morning and evening peak hours at 9:00 and 18:00 return to the traffic pattern gradually after Phase 2 opening. Although traffic flow over the weekend is much lighter than on weekdays, it also shows a 50% increase after strict restrictions are lifted. Traffic flow exiting the NUS campus is never ceased at any given time because it offers continuous access to on-campus dining centers, the University Health Center, and the National University Hospital.

The NUS hilltop location is a remote area, only accessed by maintenance and rare lab users. At any time before, during, and after lockdown, the maximum hourly traffic volume is around 10 per hour (Figure 6b). While the traffic at this site does not represent typical campus activities, this site is the most sensitive to natural events and environmental changes. The sporadic high vehicle count (bright yellow blocks) shown in Figure 6b relates directly to thunderstorms and heavy downpours, which generate large amplitudes at high frequencies that are misidentified as traffic events at this site. For example, an event stands out around 01:00 on 11 July at all four sites, denoting an obvious tropical storm event that covered the region. Similar events can be seen at other times, such as on 30 April, 21:00–22:00, on June 20 around 03:00, and more subtly on 13 June, 4:00–5:00 am. The frequency of thunderstorm events decreases from April to August 2020. Although it only takes a simple filtering operation (such as a median filter across calendar days) to remove these erratic identifications, we decide to keep them in the result to compare the effect of the environmental and natural events at different sites. Meanwhile, these erratic identifications are challenging to remove during real-time processing. We caution the users with data interpretation during strong weather events.

The access road at the Pasir Panjang Terminal Port witnesses a relatively constant flow during CB and different phases of opening (Figure 6d), serving primarily heavy container trucks that travel in and out of the port terminal. Singapore maritime ports are the only ones around the world that operate 24 h a day, seven days a week. Truck drivers, working 12-h shifts, are part of the essential workforce that sustains the lifeline between the island of Singapore and the rest of the world. As a result, the traffic volume does not vary as the government restrictions but follows the demands of international maritime transport. We observe a minimum traffic flow of around 50 motor vehicles per hour each day from 22:00 to 3:00 (Figure 7e,f), while the peak traffic flow of around 300 vehicles per hour is observed from 6:00 to 18:00 on weekdays, and from 9:00 to 12:00 on weekends. The results show that, given the choices of working shifts, significantly more drivers choose to work during daytime shifts and during weekdays.

At the West Coast Highway site, the seismometer records a traffic pattern resulting from the combined flow of both the port and the local traffic (Figure 6c). The maximum traffic volume is almost twice as large as the NUS exit site and the Pasir Panjang Terminal Port site. There is a 10% increase in the daytime hourly traffic volume after the lockdown restrictions are lifted.

From the ambient noise data recorded on the seismometer inside the West Coast Park, we investigate the frequency of urban runners and understand their workout preferences (Figure 8). We observe two consistent peaks in the running events, one right after sunrise (around 7:00) and the other right before sunset (around 19:00), showing a combined effect of working hours and the tropical climate. Compared with the running events during the lockdown, the morning peak maintains a similar intensity, while the evening peak reduces significantly after the restrictions are lifted. In particular, the number of weekend evening runners decreased by around 30% (Figure 8d). This result is markedly different compared to the Google Mobility Report for Parks (Figure 8c), as the Google data suggest that general park usage increases after Phase 2 opening and almost returns back to the before-pandemic level. In contrast, our algorithm extracts signals generated from running, which is a high-intensity physical activity. Combining the Google Mobility Report and our data, we conclude that the lockdown measure promotes participation in outdoor high-intensity physical activities, particularly on weekend evenings. As the restrictions lessen and other forms of entertainment are allowed, running quickly loses its popularity among urban residents.

## 4. Discussion

The published geophysical methods for urban monitoring are based on energy (e.g., [7,8,14]), which naturally are averages over kilometers when the frequency content is in the range of 5–20 Hz. They also do not have a resolution for the type of activities, based on energy only. Other urban-activity-monitoring methods, such as Google, Apple, or Facebook activity reports, have to anonymize the data by aggregating a large number of track events, which requires averaging over large spatial (same categories of places, for example) and temporal (daily statistics) ranges. Therefore, compared to both geophysical and existing activity tracking methods, our study provides urban monitoring solutions with unprecedented spatial and temporal resolution.

To suppress the spread of infectious diseases such as COVID-19, strict lockdown measures have to be implemented to reduce human contact around the communities where the diseases are actively propagating. The effectiveness of such measures rely entirely on compliance with public health policies. If the measures are not fully respected, the reproduction rate of the virus will not be sufficiently reduced, resulting in even longer periods of mobility restrictions. However, prolonged lockdowns could trigger adverse effects and become unsustainable both economically and psychologically [29]. Hence, a real-time understanding of policy conformance plays an extremely important role in epidemiology to strike an impossible balance between public health and individual freedom during a pandemic outbreak. Conventional surveillance solutions based on location-tracking devices, such as cameras and cell phones, evoke significant privacy concerns among the general population. Therefore, our study, based on anonymous ambient vibration data measured on nodal seismometers, provides a monitoring solution with an unprecedented spatial and temporal resolution. The resulting data are crucial for a quantitative compliance evaluation, enforcement optimization, and dynamic policy-making [30].

Activity data extracted at the Pasir Panjang Terminal Port can be used as a reference for the essential service sectors, particularly the construction sector in Singapore, whose workload is not reduced, but increased during the second half of the lockdown. The COVID-19 pandemic, as an extreme stress test, exposes the weakest links of urban systems. In Singapore, more than 95% of the confirmed infection cases in Singapore are migrant workers in the construction sector [31]. Such severe disparity in the impact of a pandemic should sound an alarm to the governments and policymakers to design better shift arrangements and provide better personal protection equipment for these essential workers.

Due to the unique geographical location and the severe demand of the highly urbanized society of Singapore, the traffic flow estimated at the Pasir Panjang Container Terminal Port also provided a glimpse of the international supply chain during the global pandemic COVID-19. Despite the worldwide impact of the pandemic and many concerns about the disruption to the global food security [32], international trade and supply chains were not immediately broken for the essentials, such as food, energy, and medical supplies. This demonstrated continuous global connectivity and support under such an intense stress test, which made a land-scarce country like Singapore functional during the pandemic [33].

Although CB disrupted the work habits of most urban residents, it did not drastically change the cycle of their recreational habits. CB in Singapore promoted evening workouts, especially during weekends. The number of evening runners slightly increased towards the later period of CB, which suggests a gradual increase in public concern towards maintaining active and healthy habits. This concern could have resulted from a gradual development of psychological fatigue from lockdown measures (e.g., [34,35]). The increased running activity stressed the importance of accessibility to open green areas in an urban environment during a time of crisis [36]. As soon as Phase 1 opening started on 1st June, the peak number of runners decreased, and the peak time of running activities shifted to a later time. In particular, a significant decrease in the Sunday evening runners was observed after Phase 2 opening, which suggests that a significant psychological strain was placed when people expect the start of a new working week. These data are paramount to sociological studies of human behavior in order to support government mitigation policy and individual response to the pandemic [37,38,39].

City management may also be interested in tracking the number of walking pedestrians in a park. A walking pedestrian, in contrast to a runner, has a lower cadence and generates a lower mechanical impact on the ground. Hence, lower seismic energy from a walker will be recorded by the sensors, making its detection a more challenging task in a noisy outdoor environment. Although the principle for walking pedestrian detection is the same as runner detection, more experiments and validations are needed for outdoor applications.

## 5. Conclusions

Urban activities cause minute vibrations in the Earth’s surface that highly sensitive seismometers can detect. The seismological data, consequently, contain rich information about urban activities that can be distilled at unprecedented spatial and temporal resolutions using our signal-processing methods. Using a nodal seismometer array, we quantify the impact of pandemic mitigation measures on the social and economic activities in different sectors of a heterogeneous urban society.

Many applications may stem from the current study, which greatly complements the existing energy-based seismic monitoring methods. After identifying the activities within the close vicinity of the sensors, signal processing algorithms can be designed to remove these known events, mine deeper into the vibration signals, and eventually attribute all possible sources of the vibrations. Such sensing and signal processing techniques with communication capability will enable real-time activity monitoring for dynamic urban management and many other smart city applications.

## Figures and Tables

**Figure 1 sensors-23-01322-f001:**
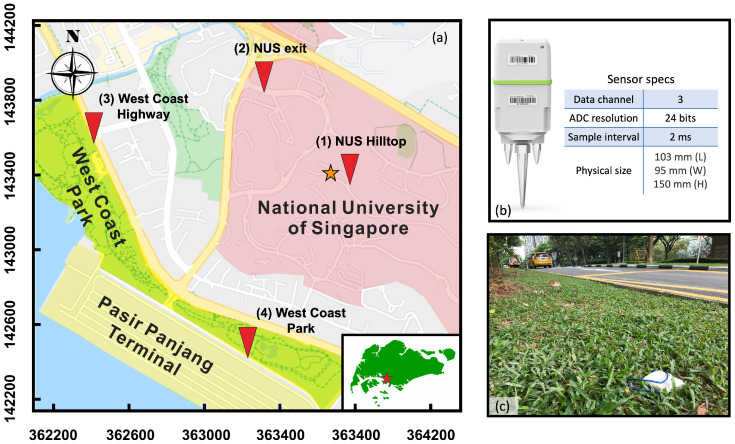
(**a**) Map of the nodal seismometer array deployed across the study area. The triangles denote the locations of the nodal seismometers and the star denotes the location of the weather station. The sites are named (1) NUS hilltop, (2) NUS exit, (3) West Coast Highway, and (4) West Coast Park, respectively. All four sensors are placed within a one-kilometer radius. (**b**) Picture of the Smart Solo IGU-16 3C nodal seismometer and specs of the sensor. (**c**) A picture of the sensor deployment in the turf grass by the road.

**Figure 2 sensors-23-01322-f002:**
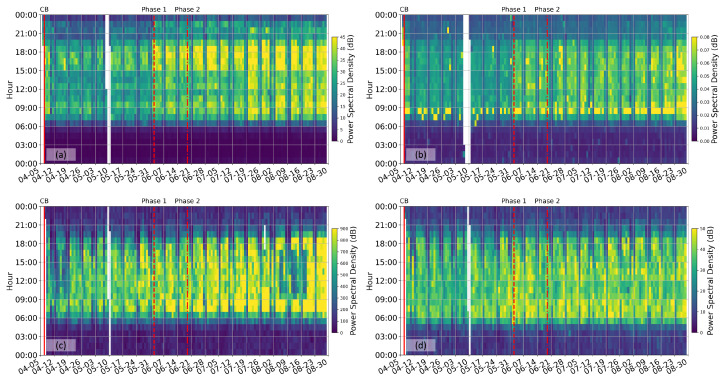
Cumulative power spectral density (PSD) between 1 Hz and 30 Hz at (**a**) NUS hilltop, (**b**) NUS exit, (**c**) West Coast Highway, and (**d**) West Coast Park, respectively. The solid red line denotes the beginning of the Circuit Breaker (CB). The two dashed red lines denote the onset of Phase 1 and Phase 2 opening according to Singapore policy, respectively. The horizontal ticks are days in the calendar year of 2020, and the vertical ticks are hours in local time. The white stripes correspond to time periods when data were missing.

**Figure 3 sensors-23-01322-f003:**
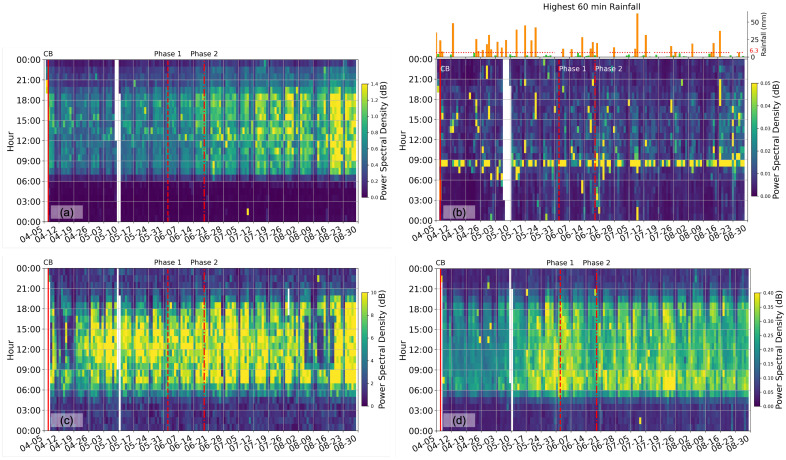
Cumulative power spectral density (PSD) between 30 and 150 Hz at (**a**) NUS hilltop, (**b**) NUS exit, (**c**) West Coast Highway, and (**d**) West Coast Park, respectively. The legends are the same as Figure 2. Rainfall data at the Kent Ridge weather station (denoted by the orange star in Figure 1) are plotted on top of (**b**), where days with intensive tropical downpours (highest 60 min rainfall greater than the mean value 6.3 mm) are plotted in orange, and the other days are plotted in green.

**Figure 4 sensors-23-01322-f004:**
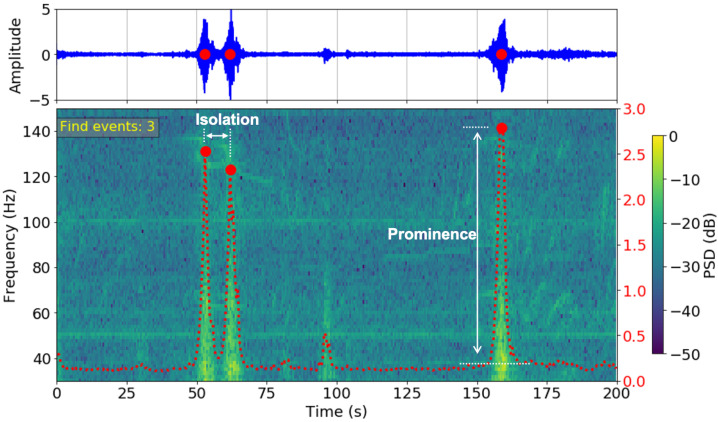
Illustration of the method for motor vehicle identification. **Top**: time domain high-frequency seismic recording. **Bottom**: spectrogram of the time domain signal. The red line is the total PSD summed over 30 Hz and 150 Hz. The large, isolated peaks are identified as motor vehicles.

**Figure 5 sensors-23-01322-f005:**
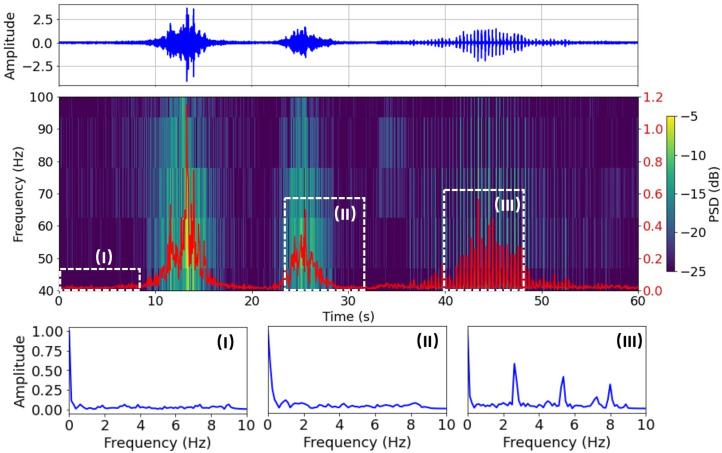
Illustration of the method for runner identification. **Top panel**: N-component seismic data filtered between 40 Hz and 100 Hz. **Middle panel**: spectrogram of the seismic data. The overlying red line is the total PSD summed between 40 Hz and 100 Hz. **Bottom panel**: normalized amplitude spectra of three 8-second segments of the PSD corresponding to background noise (**I**), motor vehicle signal (**II**), and runner signal (**III**). The runner signal is identified by strong harmonics.

**Figure 6 sensors-23-01322-f006:**
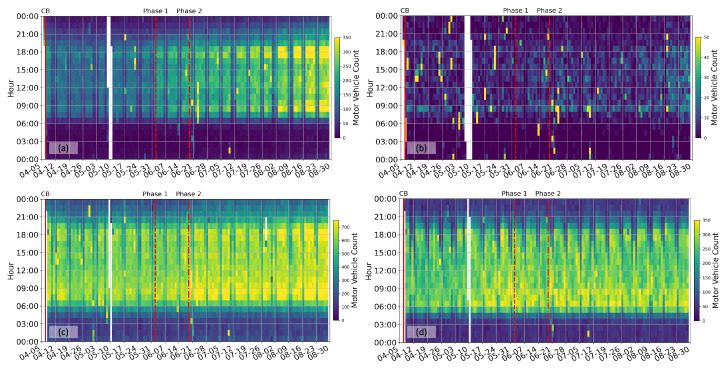
Motor vehicle count extracted from high-frequency ambient seismic data at (**a**) NUS hilltop, (**b**) NUS exit, (**c**) West Coast Highway, and (**d**) West Coast Park, respectively. The legends are the same as Figure 2.

**Figure 7 sensors-23-01322-f007:**
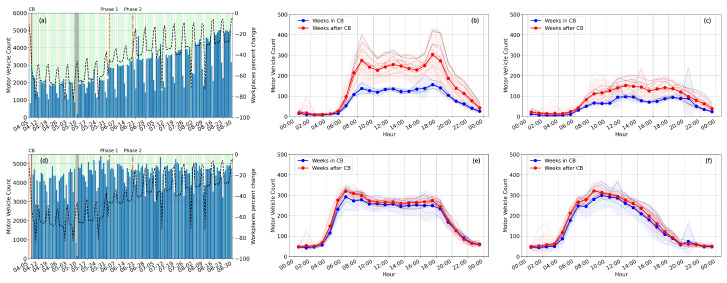
Daily and hourly motor vehicle statistics observed at the NUS exit (**top row**) and the Pasir Panjang Terminal Port (**bottom row**). In (**a**,**d**), the black dashed line denotes the percentage change at workplaces from the Google Mobility Report, while the blue bars denote the daily vehicle count derived from our data. In (**b**,**c**,**e**,**f**), vehicle counts derived from seismic data are plotted with blue and red colors to denote the periods during and after the lockdown, respectively. In (**b**,**e**), hourly motor vehicle counts are shown for the weekdays. In (**c**,**f**), hourly motor vehicle counts are shown for the weekends. In (**b**,**c**,**e**,**f**), the thick solid lines denote the average vehicle count over all the weeks during (blue) and after (red) the lockdown and the thin dashed lines denote the weekly averages. The darker dashed line denotes a later week compared to a lighter dashed line. The shaded area covers the range of twice the standard deviations above and below the mean.

**Figure 8 sensors-23-01322-f008:**
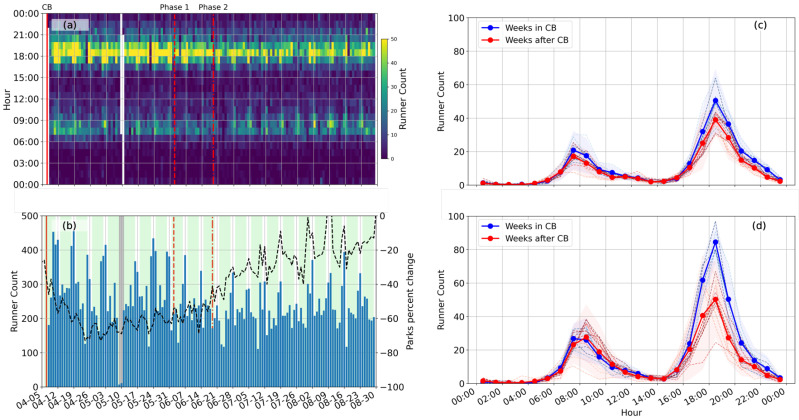
Runner count detected by the seismometer at West Coast Park. (**a**) Hourly runner count on each calendar day. (**b**) Daily runner count on each calendar day. In (**c**,**d**), runner counts are plotted with blue and red colors to denote the periods during and after the lockdown, respectively. The thick solid lines denote the average runner count over all the weeks during (blue) and after (red) the lockdown, and the thin dashed lines denote the weekly averages. The darker dashed line denotes a later week than a lighter dashed line. The shaded area covers the range of twice the standard deviations above and below the mean. In (**c**), hourly runner counts are shown for the weekdays. In (**d**), hourly runner counts are shown for the weekends.

**Table 1 sensors-23-01322-t001:** Parameters for vehicle identification at three sites.

Site	Prominence	Isolation (s)
NUS Exit	1.6	2.24
NUS Hilltop	1.6	2.24
West Coast Park	1.6	2.24
West Coast Highway	0.8	0.448

## Data Availability

The data presented in this study are available on request from the corresponding author. The data are not publicly available due to privacy concerns.

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
