# Peer review of "Quantifying Urban Activities Using Nodal Seismometers in a Heterogeneous Urban Space"

_sensors, 2023, doi:10.3390/s23031322_

Round 1

Reviewer 1 Report

Manuscript title: “Quantifying urban activities using nodal seismometers in a heterogeneous urban space”

Summary

This is a captivating manuscript that explores the growing field of urban seismology,  by using the seismic noise recorded by nodal seismometers in Singapore during the 2020 lockdowns. The authors describe a method for the quantification of urban activities (vehicle count and running activities) and claim that they achieve unprecedented spatial and temporal resolution without incurring privacy concerns regarding urban monitoring.

In my opinion, the study is relevant to the field and has good potential. However, the manuscript should be improved before publication. First of all, some figures have inferior resolution and size, and there is space for improvement in the captions, labels, and legends. The text should be better structured, clearer, and more concise. Furthermore, there are data that are used by the authors but are not well described in the text. Regarding the scientific aspect of the manuscript, my concern goes in relation to some claims on their interpretations that are not statistically supported. 

Below I detail some comments:

General comments

  • The text presents some results and discussions in the “Materials and methods section”, and introduces data (Google mobility reports, for example) in the “Results” section. I suggest the authors re-organize the text to make it more clear and more concise. Also, please expand the explanation on mobility reports (add proper citation on Google reports) and on other data that are used.

  • Often the paper citations are incomplete. If you want to cite only a few, you should add “e.g.”.

  • The authors claim that some signals are due to thunderstorm events, but they do not present any meteorological data. 

  • The authors emphasize privacy concerns but do not elaborate on the theme. 

  • Finally, the authors compare their results with Google Mobility reports, and state that they achieve “unprecedented temporal resolution”. There are many recent papers comparing Google Mobility Reports with seismic noise data, presenting correlation coefficients to quantify the similarities between the seismic and Google series, and the present work only shows some visual comparison. There are no statistical analyses to prove the authors’ claims. So I suggest expanding their interpretation.

Figures

  • Figure 1: You can make the figure bigger. It is not clear if there are two or three different colors of seismometers, maybe you should remove the alpha effect from the triangles. Why is there only one road highlighted in yellow? I suggest inserting a legend box.

  • Figures 2,3,6, and 7: the labels on the axis are too small. Please use the same font size for all figures. 

  • Figure 5: Why is the bottom panel showing a 0-10 Hz frequency band, while the signal is filtered from 40 to 100Hz?

  • Figure 6: What are these high values around 01:00, on July 11th, in the 4 stations? In 

  • Figure 7: Poor resolution. In the caption, "Data" is written with a capital letter after a comma. What is the difference between subfigures b,c,e, and f? It is not explicitly stated in the caption (and text).

  • Figures 7 and 8: What does the blue line represent? It is not expressly stated.

Specific comments

  • lines 77-78: “the government implemented a two-stage opening policy on 1st June and 19th June. Target reductions of urban mobility at 70%, 50%, and 40% at the three stages, respectively, were”: two or three stages? Opening or reduction?

  • lines 98-106: Please keep results and discussions in the results section.

  • line 110: “At this high-frequency band, ambient seismic energy shows significantly different 110 daily and long-time patterns.”: color scale limits are different between Figures 2 and 3, if you want to compare them please use the same color scale.

  •  lines 111,112: “As seismic frequency increases, the distance for seismic 111 waves to travel before their energies attenuate below an overall background noise floor 112 decreases rapidly.” : Please rephrase.

  • lines 117 and 118: Where is the information about these thunderstorms? For example, I suggest describing all data you used in the material and methods section, including weather and traffic data, and keeping the comparison analysis in another section.

  • line 156: It would be interesting if you quickly explain the reason for the harmonics pattern that runners’ signals present.

  • caption Figure 7:  “with lighter shade denoting earlier time”:  Please explain.

  • lines 184,185: “The trend of daily traffic count agreed very well with Google Mobility Report for workplaces (Figure 7a).”:  Did the authors perform any kind of quantification on this correlation?

  • lines 187,188: “We benchmark the traffic volume results with traffic data before lockdown and observe a clear reduction of 73%, 50%, and 45% during CB, Phase 1 opening,  and Phase 2 opening, respectively”:  It is unclear what the authors mean by “traffic data”. Is it the Google Mobility Report? This should be explained in the materials section. I also don’t see in Figure 7 a clear benchmark for the percentages specified. Please consider expanding this analysis.

  • lines 204-206: “Although it only takes a simple filtering operation to remove these erratic identifications, we decide to keep them in the result to compare the effect of  the environmental and natural events at different sites.” :  If the authors are willing to quantify urban activities, and claim that Figure 6 shows the Motor Vehicle Count, why keep environmental and natural events on the Figure? Please expand, or I suggest fixing Figure 6 by removing any events other than motor vehicles. -

  • Furthermore, I didn’t see until this point of the text any data about thunderstorms, although the authors claim that they appear in their signals. Please include some meteorological data to support your interpretation.

  • lines 210 and 211: “Truck drivers, working 12-hour shifts, are part of the essential workforce that sustains the lifeline between the island of Singapore and the rest of the world.”

  • lines 218 and 219: “Truck drivers, 210 working 12-hour shifts, are part of the essential workforce that sustains the lifeline between 211 the island of Singapore and the rest of the world.”: Repeated sentences.

Reviewer 2 Report

A) General remarks
The research presents in this paper a very interesting topic, as well as results that are of wider significance when it comes to the ground motion effects on the urban environment. The paper is not in every part concise and clear. The literature in the paper is adequately cited, however, some comments on the choice and significance of cited sources will be articulated in the points below.
1.     Scientific papers must be written in an impersonal form. Thus the use of personal pronouns like “we”, “our” and others should be avoided. Please carefully check the whole article.
2.    The language of the paper is adequate and no major correction to the English language is necessary. However, some sentences must be checked or divided into shorter sentences for better understanding.
3.    The small problem of the article is the description of the novelty. Please write clearly/stronger what the authors' input to the field in the case of the proposed article. This information should also be emphasized strongly in the abstract and the conclusions.
4.     The abstract is well-written. The role of the abstract is to give a basic overview of the paper. In this case, the authors give a good introduction to what the paper is about.
5.    Additionally, the authors are asked not to use conditional phrases like “can”, “could”, “may”, and “should” but only definite statements. Also do not use “weak” statements that give the feeling that the authors are not sure about the matter in question. Please check the whole article.
6.    The introduction is well-written. However, it is mostly focused on equipment and measurements. The authors are mentioning the Covid impact on urban noise but this is a part of the next chapter (materials and methods) not the introduction, which is confusing. 
7.    In the case of a review of findings on the covid impact, the review is somehow limited. The reviewer would suggest including also research on ambient noise tracking during covid not only in megacities but also in cities that are known for usual low urban noise e.g. Geneva and its surroundings DOI: 10.1515/ENG-2021-0124
8.    Unfortunately chapter 2 “Background, Materials, and Methods” is problematic to read. The following problems were detected:
a.    The background part should be a part of the introduction, not the materials and methods part
b.    The materials part is missing. No information about sensors with their parameters. No photo of the measurement setup etc. Information about signal acquisition and analysis is given after presenting some results and with not all important information on systems/tools/parameters used.
c.    In this chapter, the authors are presenting also results which is not the aim of the chapter
9.    The paper has good conclusions. However, it is suggested to state more strongly what is the novelty of the study, are next steps and the study implications.

B) Item remarks
Fig 2 and 3 are small and the text can not be read. The legend is not visible. Thus the figures are not useable to give any information to the reader.
Fig.6. is important but there is the same problem as with figures 2 and 3.
Fig. 7 no information can be read from this figure. It is out of focus, with small text, and a legend not visible. If not enlarged this figure has no use to the reader. Partially, a similar problem with Fig.8, although, due to its larger size most of it can be read.

C) Conclusions:
The article is interesting. Unfortuaneltrelly, it looks like some article writing experience (using personal pronouns, article structure, and figures preparation) is needed to improve this work. The biggest problem is the novelty statement which is weak and the chapter on materials and methods is confusing. Also, many of the figures are difficult to read. At this stage, this is borderline paper and major improvements are requested.

Reviewer 3 Report

This paper investigates the application of nodal seismometers in quantifying the urban activities such as motor vehicle movements and outdoor sport. The results can aid the urban management and policy makers to increase the efficiency of restrictions in the face of pandemics. The paper discuses valuable topic and is well written, however, the reviewer recommends that the authors provide

1-     more information on the approach taken to extract the information and omitting the noise.

2-     type of sensors (seismographs) used.

Furthermore, it is of interest that if the authors can also discuss the possibility of tracking the change in the number of people who walk at a specific path using the proposed method

Round 2

Reviewer 1 Report

Dear authors,

Thank you for addressing the issues pointed out in first review. The manuscript is much more clear now that more explanations have been added. Thank you for conducting this interesting research. I still have some comments to be considered before publication:

Abstract: “The anonymity of seismic data enabled an unprecedented spatial and temporal resolution, which potentially could be utilized by government regulators and policymakers for dynamic monitoring and urban management.”

I am still not convinced by the statement of the authors that their results present “unprecedent spatial and temporal resolution”. In fact, their own citations on urban seismology describe results with similar temporal and spatial resolution (e.g.Díaz et al, 2017). So I suggest the authors to remove this expression “unprecedent” from the abstract, or include in the discussions section why their results are unprecedent, by comparing them with the similar results cited.

Figures 2, 3 , 6, 7 and 8 should have labels with the same font size as Figures 4 and 5. I have suggested this in the first review but it wasn’t addressed. I can only read them in a big screen, or zooming the Figure.

Figure 3 – I particularly didn’t like the chosen style for the rainfall graph, but it is understandable. Please include its description on the caption properly. Why 6.3 mm is chosen as a threshold? I couldn’t find it in the text, I am sorry if it was explained. It would be interesting if the authors could include the weather station in Figure 1.

Figure 5: Why is the bottom panel showing a 0-10 Hz frequency band, while the signal is

filtered from 40 to 100Hz?

The frequency analysis shown on the bottom three panels are performed on the averaged

power spectral density in the 40-100 Hz frequency band, denoted by the red line in the middle

panel. Although the raw seismic signal is filtered to a high-frequency band, the resulting energy time series (red line) are dominated by low frequencies, depending on the type of

activity. This effect is shown in the panels on the bottom of Figure 5.

I still didn’t understand how the resulting energy of a high-passed filtered signal can be dominated by low frequencies. Could the authors clarify? Which kind of filtering was performed? If there are significant low frequencies, why doesn’t the middle panel show all frequencies from 0 to 100 Hz?

lines 184,185: “The trend of daily traffic count agreed very well with Google Mobility

Report for workplaces (Figure 7a).”: Did the authors perform any kind of quantification on

this correlation?

As we explained in the previous comment, it is trivial to compute a cross-correlation

coefficient of the two time series. The Pearson correlation coefficient is 0.368 for Figure 7a

and -0.051 for Figure 7d. However, these numbers are not informative, and we should not

imply that the Google Mobility Report should be used as a benchmark. Therefore, we choose

not to report these numbers in the paper.

Just as a matter of information, there are some other ways to quantify the similarity of two time-series with different nature (e.g. https://doi.org/10.1371/journal.pone.0253610 used CPA to check whether seismological results and Google reports had the same underlying changing points. ) I am not suggesting to perform PCA, I was just curious if the authors had any kind of quantification to support the agreement with Google Mobility Reports.

Reviewer 2 Report

Dear Authors,
Thank you for resubmitting your paper. The quality of the new version is much improved.
Some minor elements can be still improved. After the corrections, the paper will be suitable for publication.

A) General remarks
1.    Scientific papers must be written in an impersonal form…
Reviewer replay: The reviewer accepts the explanation. No additional requests.
2.    The language of the paper is adequate and no major correction to the English language is necessary. However, some sentences must be checked or divided into shorter sentences for better understanding.
Reviewer replay: The paper was improved. No additional requests.
3.    The small problem of the article is the description of the novelty…
Reviewer replay: The reviewer accepts the improved form of the novelty statement. No additional requests.
4.     The abstract is well-written…
No comments are needed. No additional requests.
5.    Additionally, the authors are asked not to use conditional phrases like “can”, “could”, “may”, and “should” but only definite statements...
Reviewer replay: The reviewer accepts the explanation. No additional requests.
6.    The introduction is well-written. However, it is mostly focused on equipment and measurements…
Reviewer replay: The reviewer accepts the improvements  
Additional comments:
The introduction, it is mostly focused on equipment and measurements. It is advised also to include some methods for analysis and long-term tracking of urban or civil engineering activities like using probabilistic methods (PPSD) DOI: 10.21008/j.0860-6897.2020.3.11 , online tools and open data access.
This element can be also used in the conclusions for possibilities of future system development.
7.    In the case of a review of findings on the covid impact, the review is somehow limited.
Reviewer replay: The reviewer accepts the improvements . No additional requests.
8.    Unfortunately chapter 2 “Background, Materials, and Methods” is problematic to read. The list of problems was provided by the reviewer.
Reviewer replay: The reviewer accepts the explanations and improvements. No additional requests.
9.    The paper has good conclusions. However, it is suggested to state more strongly what is the novelty of the study, are next steps and the study implications.
Reviewer replay: This point still requires some improvements.
Additional comments: Although the discussion part is good the conclusion chapter is missing quality. Once again please put strongly the novelty, and as this is a preliminary study of the system that probably will be developed in the future, please make it more information about future possibilities of usage, methods and analysis used etc.  

B) Item remarks
Fig 2 and 3 are small and the text can not be read. The legend is not visible. Thus the figures are not useable to give any information to the reader.
Reviewer replay: Comparing both manuscripts it looks like the quality was improved but it looks like the text was not enlarged. Please if possible make it larger for the axis caption, titles and especially the legend.
Fig.6. is important but there is the same problem as with figures 2 and 3.
Reviewer replay: similar problem. Quality ok but would suggest enlarging the font.
Fig. 7 no information can be read from this figure. It is out of focus, with small text, and a legend not visible. If not enlarged this figure has no use to the reader. Partially, a similar problem with Fig.8, although, due to its larger size most of it can be read.
Reviewer replay: the quality was improved but to read the texts and values the reader must zoom in. Maybe it would be profitable to have instead of two rows with 3 columns to have 3 rows with 2 columns. This way the graphs can be enlarged and graph a and d can be next to each other (with new numbers a and be) and then 4 other graphs. This would make the paper more esthetic and easier to follow.

C) Conclusions:
The article is interesting and was improved in most areas. After the small adjustments pointed out above the paper can be accepted for publication.
